# IoT Sensor Network Using ESPAR Antenna Based on Beam Scanning Method for Direction Finding

**DOI:** 10.3390/s22197341

**Published:** 2022-09-27

**Authors:** Md. Moklesur Rahman, Heung-Gyoon Ryu

**Affiliations:** Department of Electronic Engineering, Chungbuk National University, Cheongju 28644, Korea

**Keywords:** IoT, WSN, direction finding, BSM, ESPAR antenna, SBS, RSS

## Abstract

Wireless sensor networks (WSNs) systems based on Internet of Things (IoT) have developed rapidly in recent years. However, interference is a major obstacle to relatively long-distance communications in such networks. It is also very complicated and challenging to fix the exact location of tags in the IoT sensor networks. To overcome these problems, in this paper, an electronic steering parasitic array radiator (ESPAR) antenna used as a beamformer to handle the interference and extend the communication range from the sensors or tags is suggested. In addition, an efficient method, namely beam scanning (BS), is proposed to find the directions of tags. The beam scanning method (BSM) can be used for the selective beam switching (SBS) system by designing an ESPAR or array of ESPAR antennas with the help of CST studio. The antennas exhibit higher gain (8.17 dBi, 11.40 dBi) and proper radiation pattern at a particular direction. In addition, the MATLAB simulation findings indicate that the proposed BSM algorithm provides longer communication range, i.e., 25 m. In order to maximize range while avoiding interference, it is necessary to determine the direction and precise orientation of the tag in the WSN communication systems. Consequently, this work could be applied to an IoT sensor network such as an electrocardiogram system by providing better advantages such as higher localization accuracy and longer operating range.

## 1. Introduction

### 1.1. Motivation

Internet of Things (IoT) is mostly used in real-time electronic health, in various industries, in automation systems, and so on with intensive data transfer [1]. Nowadays, massive IoT and WSN communication with very high signal power are advancing ubiquitously [2,3,4]. Each sensor network with limited power consumption and higher efficiency becomes an important issue [5,6,7]. In addition, interference is a major obstacle to obtaining peak performance of the sensor networks [8,9,10] because the interference is exhibited within the network due to multipath fading or reflections which strongly affects the signal strength received by the radio frequency identification (RFID) reader antenna [11,12,13]. Likewise, the WSN system’s performance could degrade due to the minimum value of the signal-to-interference ratio (SIR) or the signal-to-interference and noise ratio (SINR) [14,15]. The probability of an outage or a lower level wireless connection is very high within the sensor networks for the inferior values of SIR or SNR [16,17]. To enhance the quality-of-service (QoS) within the sensor network system, the interference must be suppressed to ensure wireless connectivity. This paper presents a dense IoT approach for deploying WSN with minimum interference by designing a low-power sensor application model, i.e., ECG directional high gain ESPAR antennas. The ESPAR or ESPAR array antennas are considered one of the most advanced techniques to reduce interference and increase the capability of IoT communication channels in a massive manner because of their SBS and higher gain (11.40 dBi) [18,19,20,21,22,23,24].

In addition, there is another major problem with the location of the information signals source (RFID tags) because the antenna cannot locate the tags by itself. The location of the wireless source has drawn considerable attention in the last two decades [25,26]. Various localization methods, i.e., energy-based localization of the received signal strength (RSS) or the received signal strength difference (RSSD), allow for simple implementation compared to other traditional technologies, such as time of arrival (ToA), time difference of arrival (TDoA), direction of arrival (DoA), angle of departure (AoD), and angle of arrival (AoA) [27,28,29,30,31,32,33]. There has been progress recently toward achieving energy-based localization in various networks, including WSNs, wireless local area networks (WLANs), and automotive ad hoc networks (AANETs) [34,35,36,37,38]. It is very difficult to trace the accurate location of the data signals within the networks using ToA and TDoA. In this paper, intelligent methods, such as DoA and AoA are used to estimate the received signal power with the help of BS [39,40,41,42,43,44]. To minimize the complexities between the RFID readers and RFID tags, the BS technique is proposed. The switching beam of the ESPAR/array of the ESPAR antenna scans to find the exact location of the tags to be communicated. The scanning beams, which are open in the far field region, are produced from the SBS of the ESPAR antenna. Because the ESPAR antenna has a higher gain, it can easily achieve a longer and wider beam scanning angle with zero mean outage probability. Beam steering can be accomplished with the help of the mechanical BS because it is the prior process of SBS of the ESPAR array of ESPAR antennas. Only after fixing the position of the tags, can the SBS capture the information signal [45,46,47]. Therefore, the purpose of this research article is to determine a sensor networking system with low power consumption that can communicate over distances and find the exact location of the data signal source. The main contributions of this paper are as follows:Suggesting a low power sensor system model for ECG (electrocardiogram) application with the process of configuration.A single ESPAR antenna and an array of ESPAR which are capable of working at the ISM frequency band of (2.40~2.50) GHz antennas are designed by using CST studio.To find the exact location of tags, an efficient method, namely BS, is proposed which is also simulated with the help of CST studio.The process of evaluation based on the BSM and the range extensions are analyzed by MATLAB simulation tools.

The rest of the paper is structured as follows: Section 2 represents the WSN system model and method. Section 3 presents the BSM in a WSN system. Section 4 explains the SIR improvement and range extension. Finally, the conclusion of this manuscript is drawn in Section 5.

### 1.2. Related Works

A bean scanning angle antenna for millimeter-Wave (mmWave) sensor radar application was suggested by Yi, Z. et al. [2]. The beam scanning angle performance at 24 GHz frequency in the paper was evaluated by radar specifications such as angular distance and detections. Kuo, Y-W. et al. [3] also proposed a design of a WSN system based on the promising research topic IoT for wide area and versatile applications. This paper suggested high throughput and data rate requirements by considering the first-in-first-out (FIFO) technique at a longer battery life span. Moreover, the RFID technique is being widely and efficiently used in many IoT applications. The vital advantages of this technology are optimum operating time, low cost management, simultaneous and rapid identifications of multiple tags. Likewise, Mezzanotte, P. et al. [4] represented an RFID-based sensor network to be used in IoT applications using the concepts of radio and harmonic backscattering for wireless sensing. The reader patch antenna of the sensor network had an operating frequency of 866 MHz, and the peak gain was 5 dB.

To keep the medical hospital products or pharmaceutical entities at the best possible temperature, an IoT approach WSN system was discussed by Taillefer, E. et al. [5]. This paper aimed to propose a real-time electronic health-monitoring architecture by designing node and local management layers of the WSN. Park, J. et al. [6] developed a passive RFID-based indoor localization to better manage the interior design audits in terms of multi-stacking shelves. They expressed the idea of a reference tag that calculates the distance between an RFID reader and the reference tag to promote the accuracy of the object identification. The main contribution of this work was to determine where the product is located with economical multi-stacking-racking (MSR) as part of the warehouse management system (WMS). Moreover, Gulati, K et al. [9] presented a paper on IoT sensor networks. They suggested only a low-power data transmission technique through the sensor network’s nodes to reduce the energy consumption during the information signal receptions. In addition, Motroni, A. et al. [11] described the difficulty of understanding/determining where the pallets’ locations were being transported by forklift inside the warehouse, which is appreciated by the RFID UHF band system. The recommended forklift and sensor system enhance the habitat to the configuration of the processing unit. Specifically, the smart forklift has the ultra-wideband (UWB) tag, from which it receives the signal of UWB anchors previously installed on the warehouse site ceiling.

## 2. System Model and Method

### 2.1. Sensor System Architecture

Figure 1 shows a UHF RFID communication diagram for ECG monitoring. Although the system has not yet been implemented, this section discusses some of the potential components that might be employed to do so. These suggested components have previously demonstrated excellent performance and have been presented in many papers [3,4]. An EM4325 (2nd generation programmable IC which is compliant with ISO) made by EM microelectronics can be used to enable UHF RFID communication because in the past it has been used to download higher data transmission [1] and has been found to have a reliable measurement of data rate [2]. The EM4325 can be selected to access the RAM program via the serial peripheral interface (SPI) cable. Most companies have UHF memory entry RFID chips as a standard model. By changing the power transmission mode, the EM4325 can easily work [1,5] with the two power points used for the transmission. One depends on the power used by the reader and the other in processing which uses little energy from an outside battery [2]. The ESPAR antenna is used in this design to be integrated with the EM4325. This design system allows enough space near the antenna to provide the best possible performance. On the external side of the heart, meaning at the top of the chest, the MAX30001 microchip can be used to obtain the ECG signal due to its high performance and small size. This break/chip needs a small outer surface that makes it compact so that it can be compatible with special radios [2,4]. The TI CC2640R2 BLE microcontroller module can be used in the design as a very low-level energy consumer. For the device to function properly, power management is very important. So, it uses a small lithium polymer battery (120 mAh) in the charging and regulating process. During charging, an MCP7381T-2ACI IC microchip is applied, which allows external power to be used between the charging station and the charging system to charge up (recharge) the attached battery. This system uses the ADP1710 IC as standard which can control voltages about the amplitude of 2 V.

### 2.2. System Configuration

Figure 2 depicts the suggested system setup by taking into account a minimum interference condition as the antenna will operate with lower return losses, higher gains, and peak transmit powers. Consequently, the system is built with the ESPAR antenna. The ESPAR antenna normally functions properly in an ECG system due to the DoA beam radiation with higher gain. In the proposed configuration, the RFID tag will transmit the ECG response (signals) with the pilot vector to the antenna, and the respective antenna (ESPAR) will transmit the received signals by amplification. After processing the ECG response, it looks at the dedicated screen. This system is generally recommended as lower interference within its operating distance. Figure 2 illustrates the system configuration with multiple tags that the pilot vector sends to the central RFID reader. Each pilot has an individual ID (Identification) to carry the information from the RFID tag to the RFID reader. Each assigned tag sends information detection signals to the reader. The reader identifies each tag and assigns the tag transmission in sequence to the processor for further processing. The reader also stores the identification information, the direction (position, angle) of each tag, and the frame size in the attached memory. If the reader cannot identify the pilot, this signal will be considered as interference. In this process, a reader excludes signal interference from the system’s dedicated tags. In addition, the reader makes the ESPAR antenna receive the tag information signals with possible maximum gain. As a result, the minimum gain to interference is exhibited at a particular direction since the selective beam switching at the definite direction minimized the interference which leads the maximization of SIR. Thus, the reader sets the reactance of parasitic elements of the ESPAR antenna in order to make:Minimum gain to interference direction,Maximum gain to the authorized tags of the system.

Overall, the optimum performance could be improved by collecting data with the help of the ESPAR antenna’s directional beamforming pattern. 

### 2.3. ESPAR Antenna Design

#### 2.3.1. Single ESPAR Antenna

An ESPAR antenna consists of one active element (fed element or radiator), and the others are the parasitic elements (or parasitic radiators) attached with variable reactance. The proposed ESPAR antenna is designed using the CST studio simulator. The active monopole (#0) is placed at the center of the metal ground plane surrounded by 4 parasitic elements (#1–#4) printed on Rogers RO4725-JXR dielectric substrate as shown in Figure 3. Its relative permittivity is $\varepsilon_{r}\;$ = 2.55, and its thickness is *h* = 0.787 mm. To design a low-profile ESPAR antenna, its height has to be significantly reduced. While its radiation patterns can be successfully used to provide accurate direction of arrival estimation, a number of constructions based on the number of radiators were investigated. The distance between an active element and a passive element is λ/4 (= a). The active monopole here is fed by the coaxial connector via the central pin in order to provide 50 Ω impedance appropriately, and its height is ha = 26 mm. The parasitic elements can be opened (directors that pass through the electromagnetic wave) or shortened (reflectors that reflect the energy) to the ground by the pin diode switching circuits designed on a dielectric substrate whose height is hp = 27.2 mm. The central pin of every surrounding passive element can be connected to the ground via a corresponding switching circuit realized using inductance and reactance sets. The design parameters of the antenna are summarized in Table 1. The switching mechanism influences the passive elements’ resonance by involving a centrally located load (close to open or short circuit). As a consequence, the proposed 5-elements antenna provide $360^{{}^{\circ}}$ beam steering with each $45^{{}^{\circ}}$ discrete step using *n* = 8 directional radiation patterns. In such a setup, nth radiation pattern will have its main beam direction equal to $\varphi_{max}^{n}$ for which the radiation pattern will have to be maximum in the horizontal plane. All considered steering vectors associated with main beam directions and radiation pattern numbers, *n*, are gathered in Table 2. In addition, 3D radiation patterns of the proposed antenna during the four switching states, by which the antenna will definitely provide the strength connectivity in any network even though there is a difference in height of installed nodes, are illustrated in Figure 4. In addition, the antenna provides standard gain values which are sufficient to be used in WSN systems, IoT- based smart city deployments, and so on. Moreover, the impedance matching of the proposed antenna for the first switching state is depicted in Figure 5. The reflection coefficient below −10 dB in the considered frequency band of the antenna is presented. The minimum return loss exhibits as −24.90 dB with a bandwidth of 270 MHz, from where it could observe that the antenna exactly resonates at 2.49 GHz frequency. Comparative analysis results with some other published ESPAR antenna papers are demonstrated in Table 3 based on the radius, r (mm) of the substrate, maximum height of the antenna elements, h (mm), resonating frequency (GHz), maximum gain (dBi), and the value of return loss |S1,1| (dB) of the antenna. 

#### 2.3.2. 1 × 2 Array of ESPAR Antenna

In a 1 × 2 array of a 5-element ESPAR antenna, each ESPAR antenna is arranged in such a way that the distance between the two antennas is about one-eight wavelength, λ/8, to be operated at the ISM band frequency of (2.40–2.50) GHz. The geometrical and simulation design specification of the 1 × 2 array antenna is illustrated in Figure 6 with two single 5-element ESPAR antennas. The lumped ports of the two ESPAR antennas are setup in a different way as depicted in Figure 6.

During the 6th steering vector of the reactance set presented in Table 4, the reflection coefficient displayed in Figure 7 represents the characteristics of S1,1, S2,2, S1,2, and S2,1, as these parameters usually indicate how much power is lost or reflected from the antenna. The reflection coefficient of S1,1 and S2,2 exhibits as the value about −19.65 dB, shown in Figure 7a. The S1,2 and S2,1 return loss curves present below the −25 dB as illustrated in Figure 7b. The combined result (reflection coefficient) of the ESPAR array antenna is demonstrated in Figure 7c, and its minimum return loss value is −16.80 dB. For the other switching state, the return loss plot of the proposed array antenna maintains the same result followed by Table 4. 

The complete 3D radiation patterns of the proposed 1 × 2 array antenna are displayed in Figure 8 during the directions of $45^{{}^{\circ}}$, $90^{{}^{\circ}}$, and $315^{{}^{\circ}}$. It is observed that the peak value of gain of the proposed array antenna is 11.40 dBi. A comparative analysis result of the proposed array antenna is presented in Table 5. where the designed array antenna is compared with some published papers based on the type of array antenna, array dimensions (n × m), operating frequency (GHz), and gain (dBi).

## 3. Beam Scanning Method (BSM)

### 3.1. BSM and SBS

Different techniques could be used to find out the direction of tags; among them, beam scanning is an efficient method [1,13,15,47]. The ESPAR antenna cannot calculate the AoA of the incident beam by itself which is why it uses the beam scanning method as illuminated in Figure 9a. As the proposed system is configured with the help of 4/8-elements ESPAR/array of the ESPAR antenna, the ESPAR antenna can identify where the incident beam direction is and is the location of the interference by 8-times of radiation beam generations at each 45-degree interval. The receiver signal can be measured by using antenna beam radiation to transmit and receive at the same time. In this case, mainly the ESAPR antenna works as a receiver, and tag sensors will send the pilot signal to the ESPAR antenna with its tag ID (Identification) and communication request information. This can be seen as an incident beam, and any other signal without tag information can be considered as interference. BSM is the direction-finding method of the ESPAR antenna, which is different from the general direction finding method of phase array antennas or other multiple antennas as shown in Figure 9a. This BSM is the prerequisite process for the SBS of the ESPAR antenna in the RFID system depicted in Figure 9b. By receiving the pilot from the tag, the RFID receiver can identify the direction of each individual tag by the BSM and store the direction information in the memory of the RFID reader processor. The SBS of the ESPAR antenna in the RFID system is very useful to the RFID tag and IoT sensors for the higher SNR, SIR, and longer communication range. Using the BSM and the stored information of each individual tag, the RFID reader of the ESPAR antenna can focus the signal into the desired tag direction. Finally, the RFID reader will measure the strength of the received signal and find the tag’s location with the help of the BSM after the SBS. 

### 3.2. Direction Findings Using BSM

An ESPAR antenna is usually comprised of one active element and a number of M- passive elements maintaining an order around the active element. The active element will be connected with RF power, while the parasitic elements will be shorted with variable load reactance (x = x1, x2, x3, ……, xM). As its structure is static, it builds the next beam followed by the previous beam by shifting periodically.

Reiterating this process *M* times, *M* beams can produce M beam patterns such that it achieves the maximum beam pattern at an angle of [22,24].
(1)$$\theta_{m}=\frac{2\pi (m-1)}{M};\;\mathrm{for}\;\;m\;=\;1,\;2,\;3,\;\ldots .,\;M$$

It is salient that the omnidirectional beam pattern could be observed if and only if all the parasitic elements are shorted by the equally reactive loads approximately.

Now, the power of the radiation beam pattern at an angle, φ, on the xy-plane of the ESPAR antenna can mathematically be expressed as [24]
(2)$$P(\varphi )=C+De^{-k{\left(\frac{M\left(\varphi \right)}{\varphi_{3\;dB}}\right)}^{2}}$$
where M(φ) = ${\mathrm{Mod}}_{2\pi }$ (φ+π) − π, *k* = ln(2), $\varphi_{3\;dB}$ represents as the beam width at 3 dB, and C and D are the constant terms of the antenna.

Finally, the radiation beam pattern power for a definite area (scanning area) can be written as [20].
(3)$$P_{m}(\varphi )=P(\varphi -\varphi_{m})$$

All the passive elements of the ESPAR antenna are switched periodically in accordance with the definite interval, i.e., the harmonic beam scanning from $0^{{}^{\circ}}$ to $360^{{}^{\circ}}$ at each $45^{{}^{\circ}}$ interval. For the radiation beam pattern power $P_{m}$($\theta_{m}$) at the m-th interval, the received signal strength for our proposed system at the reader can mathematically be expressed as
(4)$$P_{reader}(\theta_{m})=S_{m}+n_{m}$$
where $S_{m}$ = $P_{m}$($\theta_{m}$), and $\theta_{m}$ are assumed as the antenna’s scanning angle at the m-th interval, and $n_{m}$ is the noise at $\chi^{2}$ distribution. In Equation (4), $S_{m}$ indicates the information of the revived signal strength based on the variation of the antenna’s scanning angle directions.

The pinnacle harmonic beam scanning angle from $0^{{}^{\circ}}$ to $360^{{}^{\circ}}$ at each $45^{{}^{\circ}}$ interval could be obtained as illustrated in Figure 10a. More specifically, the maximum scanning angle will be found out at the beam direction of $\varphi_{max}^{n}$ (i.e., $\varphi_{max}^{1}$, $\varphi_{max}^{2}$, $\varphi_{max}^{3}$,….., $\varphi_{max}^{12}$) for which the directional beam pattern will be maximum in the horizontal plane by fulfilling the switching state of the ESPAR antenna, respectively. Therefore, the peak characteristic of the received power pattern, $P_{reader}$($\theta_{m}$) for scanning angle,$\;\theta_{m}$ relative to the azimuth direction to locate the tag will be equal to the maximum directional beam angle of $\varphi_{max}^{n}$, i.e.,
(5)$${\left(\theta_{m}\right)}_{max}=\varphi_{max}^{n};\;n\;=\;1,\;2,\;\cdots ,\;8$$

Moreover, the phases and the amplitudes of all beam patterns can be analyzed at different directions. The equivalent phases among 4/8-passive elements of the ESPAR/array antenna could be obtained at a constant gradient shown in Figure 11 to be scanned by the tag at a particular direction from $0^{{}^{\circ}}$ to $360^{{}^{\circ}}$ at each $45^{{}^{\circ}}$ interval. From the below Figure 11, it is obtained that the beam gain pattern of the single ESPAR antenna is about two times lower than the array antenna for scanning.

### 3.3. BSM Operation

It is very important to have the strengthened data signal at the reader antenna from a particular direction of a tag in any WSN system. Figure 12 shows the tag’s position in different cases. To understand the BS operation, the number of parasitic elements of the ESPAR antenna is considered here as 1, 2, 3, …, *N*. Actually, the tag’s antenna sends the pilot carrier toward the ESPAR antenna which is attached to the RFID reader. In this way, the location of the tag is stored in the reader section memory. During the condition of shorting all the parasitic elements of the ESPAR antenna, the reader could easily detect the source of information signal positions with the help of RSS. When the data signal from an unknown source is coming at the ESPAR antenna and if element #1’s configuration set of the antenna is ON, the RSS by this element’s set will be the strongest as depicted in Figure 12a. In this case, the tag’s position will also be decided by element #1’s set, and the tag number will be stored in the reader data base. If the two neighboring passive element sets are shorted, the RSS will be equal at both elements. The decision of the tag’s location could be traced by both element (#1 and #N) sets as shown in Figure 12b. When the two elements #5 and #N sets configuration is closed, and the RSS at element #N is approximately two times higher than element #5, the tag’s position must be obtained by the element of #N as illustrated in Figure 12c. As the WSN system is comprised of multiple numbers of tags, there is a possibility to have different data signals presenting at the same reader ESPAR antenna’s elements. Tag collision happens when two or more tags’ individual information signal identification reaches the reader at the same time, In this case, the reader confuses the identity of the tags. To avoid tag collision, the ESPAR antenna receives one request at a time, and other pilots stay in line until the first request is completed. Only after completing the first request, will the ESPAR antenna receive the first waiting pilot. There is no way to interfere with the one pilot or others waiting because the used antenna depends totally on the switching system and can receive only one request at a time. The location of the moving tag is very difficult to find, but a directional beam antenna (ESPAR) can easily trace it. When tag 1’s information signal reaches element #1 after some time varying its location, the signal goes to element #N. The reader data set will spontaneously change until the tag’s position becomes fixed; this scenario is illustrated in Figure 12e. In the case of longer distance between the RFID reader and the RFID tag, it is very difficult to find the tag’s location until the strengthened signals come to the reader section antenna. If the RFID tag is out of range or it is affected by more interference signals, the tag’s position will not be noted by the reader as shown in Figure 12f. Therefore, there is a very low probability of collisions among the tag’s signals because the directional switching beam of the antenna only accepts signals from a definite position. In addition, the antenna will be able to handle the information signals coming from the tag source with the optimum strength by passing them to the reader. In this way, the DoA or AoA can detect the tag’s position with the help of beam scanning. As a result, the interference management and the localization of tags through the sensor system are regulated optimally. It is important to note that to increase the accuracy, multiple scannings are necessary, but they take longer. The direction findings Algorithm 1 is presented regarding RSS at different ESPAR elements.
**Algorithm 1:** Tag’s Localization Beam Scanning Algorithm**If** all elements == 0 No data signals will be received;**end****If** element #1 == 1  RSS at #1 is very high (↑)**end****If** elements #1 & #N == 1 RSS (↑) obtained at #1 or #N**end****If** elements #5 & #N == 1 RSS decided by any one of #5 and #N  **If** RSS at #N >> #5   tag identification is noted by #N  **end****end****If** element #N == 1 tag 1 and tag 2’s pilot arrived at #N **If** tag 1’s pilot is accepted  tag 2’s pilot will be in line **end****end****If** all elements == 1 Moving tag will be noted by an element**End**

## 4. SIR Improvement and Range Extension

### 4.1. SIR Improvement of ESPAR Antenna

When the interference power is two times higher than the RFID tag power, SIR will be −3 dB lower. Again, if the interference power is two times lower than the RFID tag power, SIR will be 3 dB higher [38,39]. The directional gain of the proposed single ESPAR antenna at the receiver is 8.17 dBi, and the minimum null gain of the antenna is −30 dB. These characteristics of the antenna will definitely lead to a higher value of SIR (i.e., more than 38 dB), hence increasing the range and quality of communications. Due to the array antenna, the value of SIR will be much higher than the single ESPAR antenna at the respective direction. Figure 13 depicts, nulling and interference to improve SIR value of single ESPAR antenna during $135^{{}^{\circ}}$ beam direction with the help of CST studio. 

### 4.2. Communication Range Extension 

The signals received from a particular source by the antenna will be highly sensitive and strengthen as it has optimum beam gain at all the respective directions. According to the friis’ power transmission formula, the power received by the reader antenna at a distance d from the transmitter or sensor tag antenna in an obstacle-free environment is defined as [29,35]
(6)$$P_{reader\;}\;=\;\frac{P_{tag}G_{t}G_{r}\lambda^{2}}{{\left(4\pi \right)}^{2}d^{n}P_{L}}$$
where $P_{reader}$ is the power received by the reader antenna, $P_{tag}$ is the power transmitted by the sensor side antenna, $G_{t}$ and $G_{r}$ are the tag and reader antennas gain, λ is the wavelength at the respective operating frequency, $P_{L}$ is the path loss of the system, and *n* is the path loss factor for free space its value is 2.

The numerical result is presented depending on the designed antenna gains and shadowing channel. The parameters are used for modeling the RSS relative to the distance as summarized in Table 6. The simulation result shows that the maximum operational distance for the proposed array antenna’s beam gain is more than 25 m (m), whereas the minimum distance is also higher than 22 m (m), illustrated in Figure 14. For the single ESPAR antenna gain 8.17 dBi, the distance is also higher than 17 m. Compared to the previously published papers beam gains, such as 7.99 dBi [13] and 6 dBi [24], the RSS result is also depicted in Figure 14. It is shown that the distance due to the lower beam gain is less than the distance obtained due to the higher gain proposed antennas.

A comparative analysis and drawback of the previous papers compared to this article is depicted in Table 7 depending on key factors, such as system model of the sensor network, antenna design, localization technique, analyzing SIR/RSS, and the communication range (m). The localization techniques to find the tags are not highly efficient, and the distance of operation is very poor. These are the major limitations of the previously published papers.

## 5. Conclusions

This proposed system incorporates dense information processing in a new massive UHF RFID IoT system using BSM to localize the tags and investigating the link budget to estimate the distance between the RFID readers and the RFID tags to monitor the heart’s activities by suppressing interference with the help of an ESPAR/array of ESPAR antennas. The BSM was used to fix the exact location of the RFID tags along with its particular direction. The accuracy of the BSM for shorter distance (≥25 m) communication is very high compared with other methods. The beam steering of the ESPAR or array of ESPAR antennas exhibits very good performance regarding radiation patterns, gains (8.17 dBi, 11.40 dBi), and the reflection coefficients. Thus, the proposed design could be considered to be a reliable, robust, and low power IoT sensing system.

## Figures and Tables

**Figure 1 sensors-22-07341-f001:**
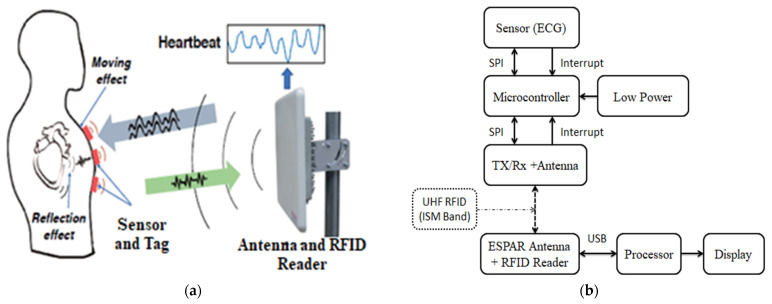
(**a**) Illustration of the working principle of ECG; (**b**) UHF RFID communication system for ECG monitoring.

**Figure 2 sensors-22-07341-f002:**
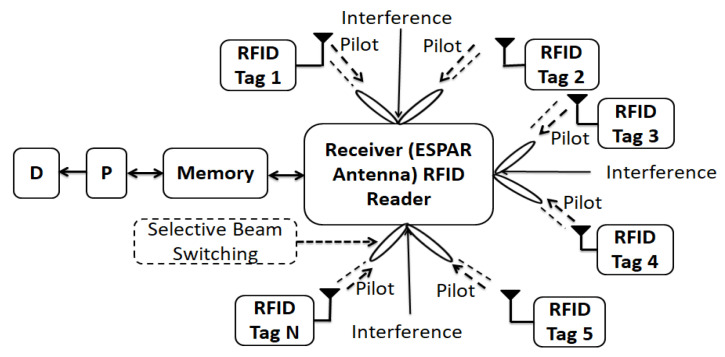
System configurations with central reader; D: display, P: processor.

**Figure 3 sensors-22-07341-f003:**
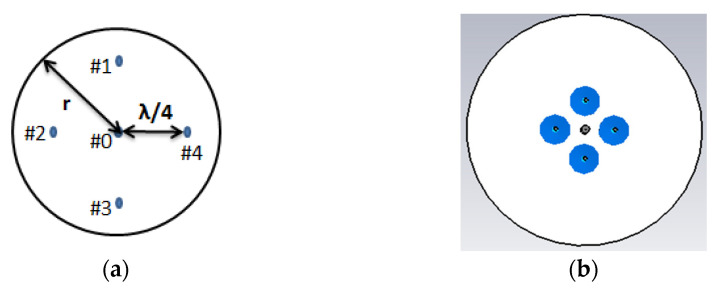
(**a**) Geometry of the proposed ESPAR antenna; (**b**) simulation view of the proposed ESPAR antenna.

**Figure 4 sensors-22-07341-f004:**
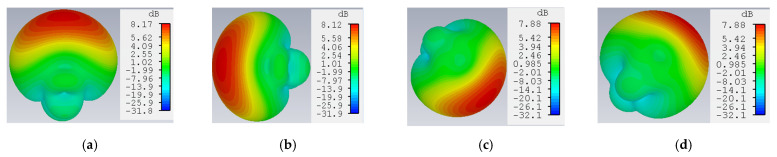
3D radiation patterns of the antenna at: (**a**)$\;0^{{}^{\circ}}$; (**b**) $90^{{}^{\circ}}$; (**c**) $225^{{}^{\circ}}$; (**d**) $315^{{}^{\circ}}$.

**Figure 5 sensors-22-07341-f005:**
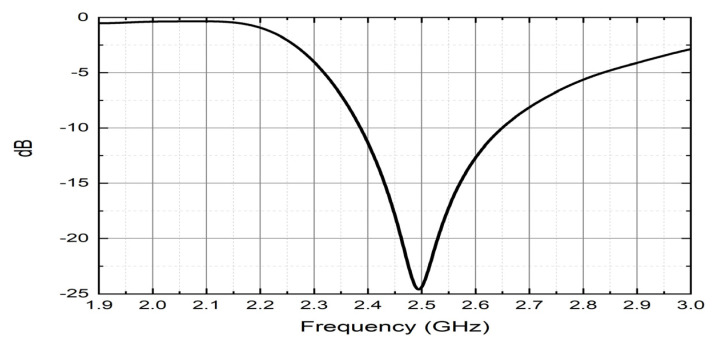
Reflection coefficient of the antenna.

**Figure 6 sensors-22-07341-f006:**
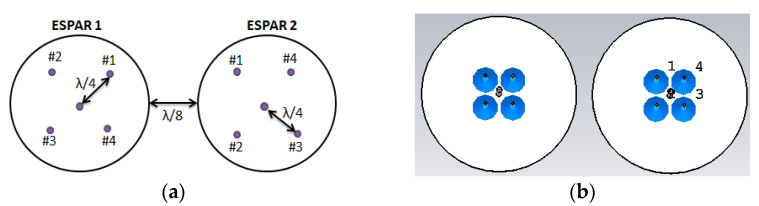
(**a**) Geometry of the ESPAR array; (**b**) simulation view of the ESPAR array antenna.

**Figure 7 sensors-22-07341-f007:**
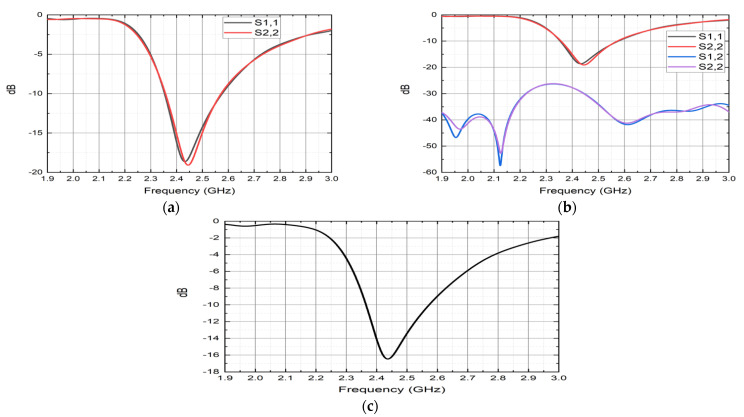
(**a**) Reflection coefficients of S1,1 and S2,2; (**b**) complete reflection coefficient during $\varphi_{max}^{6}$ = $225^{{}^{\circ}}$; (**c**) reflection coefficient after far-field combination.

**Figure 8 sensors-22-07341-f008:**
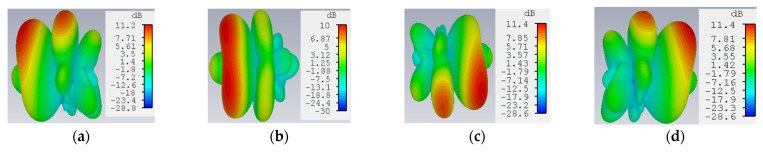
3D radiation patterns of the antenna at: (**a**)$\;45^{{}^{\circ}}$; (**b**) $90^{{}^{\circ}}$; (**c**) $225^{{}^{\circ}}$; (**d**) $315^{{}^{\circ}}$.

**Figure 9 sensors-22-07341-f009:**
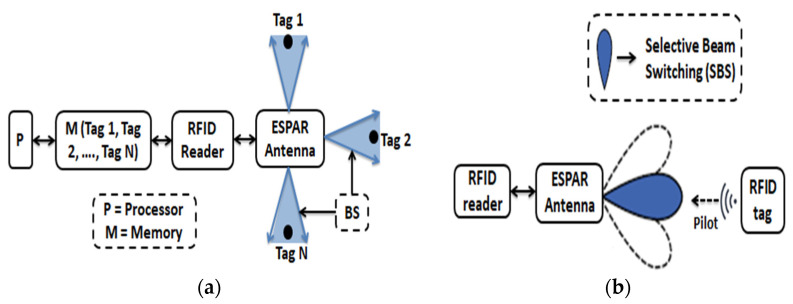
(**a**) BSM for the RFID sensor networks; (**b**) SBS at a particular direction of the RFID tag.

**Figure 10 sensors-22-07341-f010:**
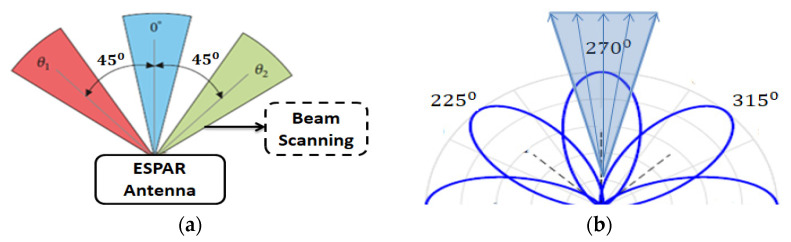
Beam scanning at different directional angle: (**a**) BS of the ESPAR antenna at respective directions; (**b**) BS at $270^{{}^{\circ}}$ followed by angular beam direction at each $45^{{}^{\circ}}$ interval.

**Figure 11 sensors-22-07341-f011:**
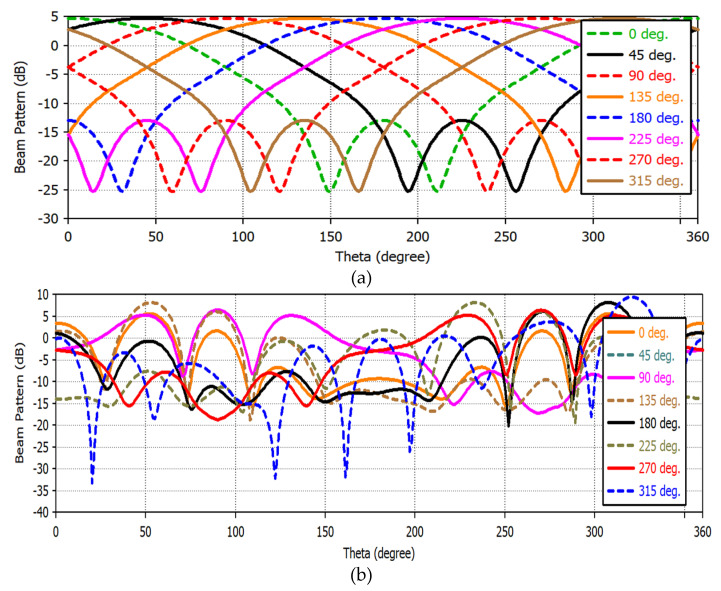
(**a**) Beam scanning of the single antenna based on beam directions; (**b**) beam scanning of the array antenna based on array beam directions.

**Figure 12 sensors-22-07341-f012:**
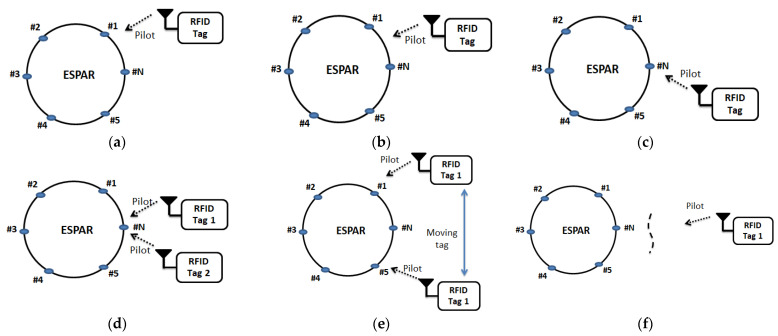
Passive elements of the ESPAR antenna decide the tag’s direction: (**a**) passive element #1 decides the tag’s direction; (**b**) decision of the tag’s position is selected by passive elements #1 or #N; (**c**) tag position is decided by element #N set configuration; (**d**) two pilots of the separate tags coming toward the parasitic element #N; (**e**) the tag’s location decision during moving; (**f**) when the tag is farther from the ESPAR antenna or due to the obstacle, the RSS at the parasitic elements is affected by noise; (1, 2, 3, …, N are the parasitic elements of the ESPAR antenna).

**Figure 13 sensors-22-07341-f013:**
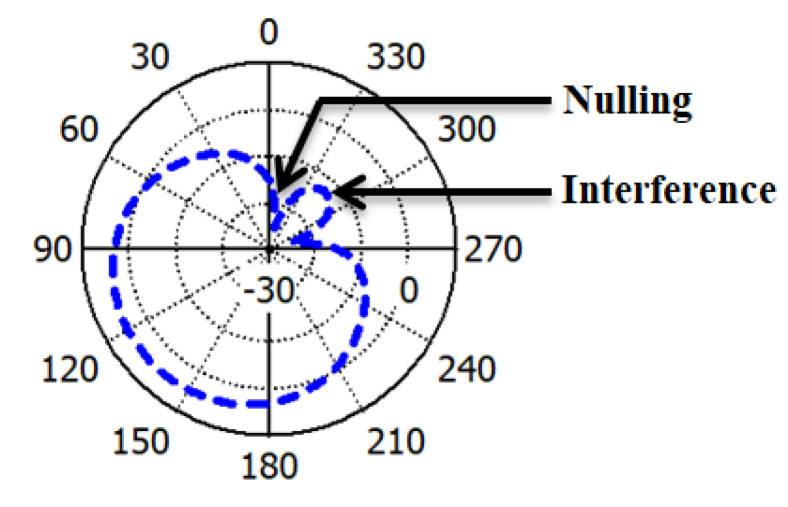
Polar radiation pattern of the single antenna by indicating nulling and interference.

**Figure 14 sensors-22-07341-f014:**
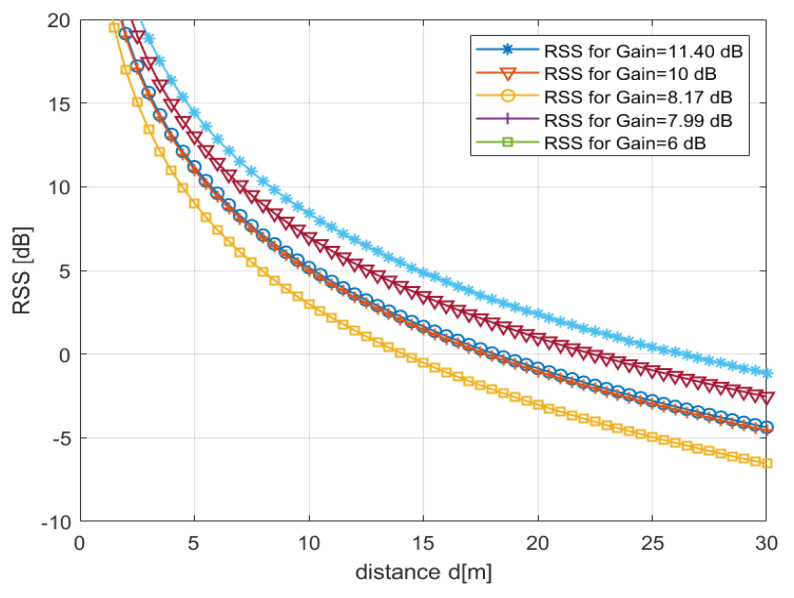
RSS over the distance based on the antenna gains.

**Table 1 sensors-22-07341-t001:** ESPAR antenna design parameters.

Parameters	Value
Radius of the substrate, r	90 mm
Height of the active element, ha	26 mm
Height of the passive elements, hp	27.2 mm
Distance between active and parasitic elements, a	λ/4
Nominal impedance	50 Ω
Operating frequency, f	2.49 GHz

**Table 2 sensors-22-07341-t002:** Reactance set of the single ESPAR antenna.

Number, *n*	$\mathrm{Directions},\;\varphi_{max}^{n}$	Lumped Ports			
		#1	#2	#3	#4
1	$0^{{}^{\circ}}$	0.1 pF	0.5 nH	0.5 nH	0.5 nH
2	$45^{{}^{\circ}}$	0.1 nH	0.5 pF	0.1 pF	0.7 nH
3	$90^{{}^{\circ}}$	0.5 nH	0.1 pF	0.5 nH	0.5 nH
4	$135^{{}^{\circ}}$	0.5 nH	0.1 nH	0.5 pF	0.1 pF
5	$180^{{}^{\circ}}$	0.5 nH	0.5 nH	0.1 pF	0.5 nH
6	$225^{{}^{\circ}}$	0.5 pF	0.1 nH	0.7 nH	0.1 pF
7	$270^{{}^{\circ}}$	0.5 nH	0.5 nH	0.5 nH	0.1 pF
8	$315^{{}^{\circ}}$	0.5 pF	0.1 pF	0.1 nH	0.7 nH

**Table 3 sensors-22-07341-t003:** Comparative analysis of the single ESPAR antenna.

References	Substrate Radius, r (mm)	Max. Height, h (mm)	Operating Frequency, f (GHz)	Peak Gain (dBi)	|S1,1| (dB)
[13]	60.34	30.34	2.48	7	N/A
[14]	246	71	0.87	6.99	−12.2
[15]	60	30	2.50	5.67	−11
[18]	120	30.60	2.40	5.70	−15
[19]	N/A	40.5	2.20	7.73	N/A
[21]	56	N/A	3.50	7	−19
[22]	400	200	0.60	9	N/A
[23]	121.2	N/A	2.45	5.21	N/A
[30]	140	26	2.40	8.44	−16
This	100	27.2	2.49	8.17	−24.90

**Table 4 sensors-22-07341-t004:** Reactance set of the proposed 1 × 2 ESPAR array antenna.

$\mathrm{Directions},\;\varphi_{max}^{n}$	Lumped Ports							
	ESPAR-1				ESPAR-2			
	#1	#2	#3	#4	#1	#2	#3	#4
$0^{{}^{\circ}}$	0.15 pF	0.45 nH	0.45 nH	0.45 nH	0.2 pF	0.45 nH	0.45 nH	0.45 nH
$45^{{}^{\circ}}$	0.45 nH	0.2 pF	0.45 nH	0.45 nH	0.3 pF	0.45 nH	0.45 nH	0.45 nH
$90^{{}^{\circ}}$	0.45 nH	0.1 pF	0.1 pF	0.45 nH	0.1 pF	0.2 pF	0.45 nH	0.45 nH
$135^{{}^{\circ}}$	0.45 nH	0.45 nH	0.2 pF	0.45 nH	0.45 nH	0.2 pF	0.45 nH	0.45 nH
$180^{{}^{\circ}}$	0.45 nH	0.45 nH	0.45 nH	0.15 pF	0.2 nH	0.2 pF	0.45 nH	0.45 nH
$225^{{}^{\circ}}$	0.45 nH	0.45 nH	0.45 nH	0.15 pF	0.2 nH	0.45 nH	0.2 pF	0.45 nH
$270^{{}^{\circ}}$	0.1 pF	0.45 nH	0.45 nH	0.1 pF	0.45 nH	0.45 nH	0.1 pF	0.1 pF
$315^{{}^{\circ}}$	0.2 pF	0.45 nH	0.45 nH	0.45 nH	0.45 nH	0.45 nH	0.45 nH	0.3 pF

**Table 5 sensors-22-07341-t005:** Comparative analysis of the proposed array antennas.

Ref.	Array type	Array (n × m)	Frequency (GHz)	Gain (dBi)
[27]	Patch	1 × 8	1.80	7.32
[30]	Patch	2 × 1	2.40	9.22
[38]	Patch	1 × 8	28	6.99–10
[40]	Patch	1 × 2	5.80	9.19
[41]	Patch	2 × 2	24	8.5
This	ESPAR	1 × 2	2.40–2.50	9.95–11.4

**Table 6 sensors-22-07341-t006:** Parameters for RSS.

Parameters	Value
Frequency	2.45 GHz
Tag antenna power, $P_{tag}$	25 dB
Reader antenna gain, $G_{r}$	10 dB, 11.40 dB
Tag antenna gain, $G_{t}$	8 dB
Path loss, $P_{L}$	−2 dB
Path loss factor, *n*	2

**Table 7 sensors-22-07341-t007:** Comparisons with some published papers.

Related Works	Sensor System with Antenna	Localization Technique	SIR/RSS Analysis, Range (m)
Mezzanotte, P. et al. [4]	Sensor system with angular slot patch antenna	No	Yes, the peak operating range is 5 m
Ding, W. et al. [13]	Sensor application only	Non-convex position	Yes, not specified
Zhang, Y. et al. [15]	Sensor application only	Linear technique	Yes, no estimated range
Xu, B et al. [16]	Only sensor system model	No	Yes, range is about 20 m
Mekelleche, F. et al. [18]	Sensor model only	Trilateration, Bounding Box, APSAoA	No
Ying, J. et al. [25]	No	CRLB technique	Yes, operating range is 15 m
Kumar, D. et al. [26]	Yes	No	Yes, opting range is 25 m
A. Alanezi et al. [29]	No	MVPA algorithm	Yes, the range is 4.3 m
Khosla, D. et al. [32]	Yes	No	No
Wang. Q. et al. [33]	Yes	Relative	RSS relatively compared
Benes, F. et al. [34]	Yes	No	Yes, the range is 10 m
This proposed	Yes	BSM	Yes, range more the 25 m

## Data Availability

Not applicable.

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
