# Peer review of "IoT Sensor Network Using ESPAR Antenna Based on Beam Scanning Method for Direction Finding"

_sensors, 2022, doi:10.3390/s22197341_

Round 1

Reviewer 1 Report (Previous Reviewer 2)

This manuscript is the revised version of the previous manuscript that I served as a reviewer. The quality of this revised manuscript is much better than the previous one. I have the following comments for the authors to improve the clarity and readability of this manuscript further.

1. The Abstract part needs to be improved. What are the relations between fixing the exact location of the tag, handling the interference, and extending the communication range? Why do we need to fix the exact location of tags in the IoT sensor networks?

2. In subsection 2.1, the authors listed the names of real components in the UHF-RFID communication system. I wonder how the authors realized these components in MATLAB?

3. Line 157: the terminology "interference free-state" is confusing. Please clarify it.

4. Line 262-263: What are S1,1, S2,2, S1,2, and S2,1? The authors should define them before usage.

5. Algorithm 1 should be presented in a more general and concise form.

6. There is still a lack of comparing the performance of the proposed localization technique with the previously proposed techniques in the literature by using numerical results in several scenarios. The authors should add this information to convince the readers of the advantage of the proposed localization technique.

7. Many small typos and notation issues can be found throughout the manuscript. I suggest using some checking tool such as Grammarly to fix these issues.

8. Illustrative and numerical result figures are not presented in high-resolution forms. I recommend using eps files instead of jpeg files to enhance the quality of figures after converting the whole manuscript file to pdf.

Author Response

Thank you very much for your valuable time, enrich suggestions, important comments and questions. We have taken all comments, suggestions and questions carefully. Changes and improvements made to the revised manuscript are summarized below submitted word file.

Reviewer 2 Report (New Reviewer)

The manuscript is well written and presented even though it doesn’t follow the traditional Intro, Material and Method, results, discussion and conclusion style. I find the manuscript qualified for publication at Sensors at MDPI provided the following minor issues are addressed:

1-     Some lines look different in font size than others. For example, lines 11-13.  Please check this for the entire manuscript

2-      Line 59 : please rewrite this: “It is very default”

3-     I am not sure referring to authors like ABC, D. et al is correct. Usually, we use ABC et al. Please check in the entire manuscript

4-     Line294-295 : Please rewrite this sentence. Also needs reference: Different techniques could be used to find out the location of the tags among them, BSM is the best.  

-        It is not clear what does the green highlight represent in the text?

-

Author Response

Thank you very much to the reviewers for your valuable time, enrich suggestions, important comments and questions. We have taken all comments, suggestions and questions carefully. Changes and improvements made to the revised manuscript are summarized below submitted word file.

Reviewer 3 Report (New Reviewer)

The paper titled "IoT Sensor Network Using ESPAR Antenna Based on Beam 2 Scanning Method for Direction Finding" presents an interesting antenna structure for direction finding applications. Some of the suggestions to improve the paper are as given below:

1. The gain values mentioned in line 19 of the Abstract are ambiguous. It would be better to write two values separated by a comma rather than a slash symbol. Also, the unit of the gain should be in dBi.

2. In the abstract, in line 24, the authors have mentioned "massive IoT networks like ECG systems". But an ECG system uses 12 lead or nodes which is not a huge number to call it a massive  IoT network. Authors can use an appropriate word in place of the massive network.

3. On page 3, line 128, the sentence is not making sense. Rewrite the sentence.

4. The peak gain unit mentioned in Table 3 should be in dBi.

5. Will the passive parasitic elements near the active element affect the radiation pattern as they are of almost the same length and are conductors?

6. Why authors chose to represent 4 directional radiation patterns in 2D form and the rest 4 in 3D form? For example in Figures 4 and 5.

7. There are no measurement results presented after developing the antenna and testing it with a network analyzer and in an Anechoic chamber. Can the authors justify as to why there are only simulated results?

Author Response

Thank you very much to the reviewers for your valuable time, enrich suggestions, important comments and questions. We have taken all comments, suggestions and questions carefully. Changes and improvements made to the revised manuscript are summarized below submitted word file.

Round 2

Reviewer 1 Report (Previous Reviewer 2)

The authors have addressed all my previous comments to improve the quality of the manuscript. I really appreciate the authors' efforts and strongly recommend this manuscript be published in its current form.

Reviewer 3 Report (New Reviewer)

The authors have answered my comments and have incorporated changes wherever required. The manuscript looks fine to me.

This manuscript is a resubmission of an earlier submission. The following is a list of the peer review reports and author responses from that submission.

Round 1

Reviewer 1 Report

The paper presents a simulation study on ESPR antenna based IoT sensor network. Some of the discussion regarding the antenna design are interesting. The topic of the paper is suitable for MDPI Sensors journal. However, there are some issues with the paper that need to be addressed. The writing needs to be improved. In fact, there are several sentences that are not very coherent. The paper is also rather lengthy. This should be significantly reduced to highlight the main contributions of the paper. My detailed comments are listed below:

  1. Line 11, in abstract “But interference is a vital issue to be communicated….”. This sentence is incoherent. It is not clear what the authors are trying to imply. Please revise.
  2. Through lines 60 to 68, the full forms of the terms DoA, AoA, ToA etc. have been defined twice. Please correct.
  3. The introduction, related works and motivation should be confined into a single introduction section. These sections span 3 pages currently. The actual information there can be condensed to around 2 pages. For example, the literature review can be much more succinct. Please revise and make a succinct introduction.
  4. When talking about interference in wireless network, the authors may wish to provide some reasons for the interference. For example, multi-path propagation and Rayleigh fading. A few relevant papers should be cited. For example:
  5. https://doi.org/10.1145/1143549.1143757
  6. https://doi.org/10.1109/SIBIRCON.2010.5555104
  7. In line 104-105, it is not clear what is meant by the sentence “The reader antenna 104 of the sensor network is connected with a patch antenna..”. Please clarify.
  8. Through line 187 to 205, the authors mention the model number of several hardware. However, as far as I can tell, the paper only discusses simulation results. If that is the case, then what is the relevance of the hardware information? It can easily mislead a reader to think that the paper contains experimental data. If the authors want to discuss the feasibility of their method, they can write this in a much more clear and succinct way. If indeed some experiment was performed, please provide experimental data.
  9. In line 218-219, “This system is generally recommended for optimal distance, as interference is considered to be very close to zero” is very unusual. What do the authors mean by “recommended for optimal distance”? What is optimal distance? By definition of optimum, any system is recommended for optimum configuration.
  10. In figure 4, instead of the 3D color plots, please provide a standard 2D polar plot of the radiation pattern (similar to Fig 12). The results would be more quantitative and easy to read that way. Please do the same for other similar figures.
  11. In line 500-501, the equation is often referred to as the Friis power transmission formula. Please mention the name of the equation in the paper.
  12. Fig 10 does not have any legends. It is not clear what the different colors represent.
  13. It does not appear that the entire system has been simulated. Only parts of the system (antenna, beam scanning etc.) have been separately (and independently) simulated. It should be noted that the system performance is often more complicated than the performance of the individual components. Please comment on this in the manuscript.
  14. My main criticism of the paper is that the contributions of the paper are not well discussed. At the introduction section (and partially in the abstract), the authors should clearly say what they have done in the paper. For example, “we simulate an antenna using software… and then simulate the beam scanning technique using ….”. Please modify the paper to address this.
  15. Was any simulation performed regarding the sensor network? Or is the work limited to the antenna simulation? Please clarify.
  16. How is Fig. 9(b) generated? Is it simulated? If so, explain.
  17. As far as I can tell, beam scanning was not simulated even though there is a section on it (section 4.1.)

Reviewer 2 Report

This paper utilizes an array ESPAR antenna to improve the efficiency of direction finding and to extend the communication range in the IoT sensor networks. In general, the idea is valid, however, the organization and numerical results of this paper need to be significantly improved. Specifically,

1. There are many typos and the writing of this paper is poor. The abbreviations BSM appear twice in the abstract. There is no need to use uppercase letters for the full names of abbreviations. Hyphens need to be added in several words such as "signal-to-noise ratio". The more quantitative findings should be given in the abstract part instead of qualitative comments. Notations in Table 1 should include subscripts.

2. The authors should avoid splitting subsection 2.1 into many paragraphs. Instead, it should be presented with idea connections. The evaluation of the proposed array ESPAR antenna in sensor networks is based on simulation. This should be clarified in the abstract and motivation part to avoid confusion with real implementation and measurement.

3. In subsection 4.1, the authors mentioned that the 13-element ESPAR antenna was configured in the sensor network. However, the attributes (3D radion pattern, reflection coefficient) of this array antenna were missing. The author should replace the attributes of the 2-element ESPAR antenna with those of a 13-element ESPAR antenna.

4. Some paragraphs and equations in subsection 4.2 should be moved to subsection 3.3 as they still mention the antenna's properties. The pseudo-code of Algorithm 1 is not generic because if the number of tags/antenna elements is high, then the length of this algorithm becomes very long.

5. Numerical results are very poor. There is only one figure for RSS versus distance. No comparisons with other previously proposed antennas were added. The reviewer suggests the authors provide more evidence for claiming that the proposed array antenna brings benefits to the direction finding and communication range in IoT sensor networks.